# The Association of Grip Strength with Depressive Symptoms among Middle-Aged and Older Adults with Different Chronic Diseases

**DOI:** 10.3390/ijerph17196942

**Published:** 2020-09-23

**Authors:** Priscila Marconcin, Miguel Peralta, Gerson Ferrari, Margarida Gaspar de Matos, Margarida Espanha, Eugenia Murawska-Ciałowicz, Adilson Marques

**Affiliations:** 1Faculdade de Motricidade Humana, Universidade de Lisboa, 1649-004 Lisboa, Portugal; margarida.gaspardematos@gmail.com (M.G.d.M.); mespanha@fmh.ulisboa.pt (M.E.); 2CIPER, Faculdade de Motricidade Humana, Universidade de Lisboa, 1649-004 Lisboa, Portugal; mperalta@fmh.ulisboa.pt (M.P.); amarques@fmh.ulisboa.pt (A.M.); 3ISAMB, Faculdade de Medicina, Universidade de Lisboa, 1649-004 Lisboa, Portugal; 4Laboratorio de Ciencias de la Actividad Física, el Deporte y la Salud, Facultad de Ciencias Médicas, Universidad de Santiago de Chile, USACH, Santiago 7500618, Chile; gerson.demoraes@usach.cl; 5Department of Physiology and Biochemistry, University of Physical Education, 51-612 Wroclaw, Poland; eugenia.murawska-cialowicz@awf.wroc.pl

**Keywords:** handgrip strength, depressive symptoms, chronic disease, SHARE

## Abstract

Low grip strength has been associated with an increase in depressive symptoms, independent of age group or gender, although the literature has not investigated this association among different chronic diseases. The present study aims to investigate the association of grip strength and depressive symptoms among middle-aged and older adults with different chronic diseases. A cross-section of data from the Survey of Health, Ageing, and Retirement in Europe wave 6 (collected in 2015) was analysed. Grip strength was measured by a handgrip dynamometer, and the European Depression Symptoms 12-item scale (EURO-D) was used to assess depressive symptoms. Multivariable logistic regression analysis was conducted. Those in the high strength tertile had 42% (95% confidence interval: 0.50, 0.71; *p* < 0.005) and 41% (95% confidence interval: 0.50, 0.70; *p* < 0.001) lower odds of depressive symptoms in the ‘no disease’ and in the ‘metabolic diseases’ groups of participants, respectively, compared with those in the lower strength tertile. No statistically significant relationship between grip strength and depression was observed in the ‘arthritis diseases’ group of participants. The association of grip strength with depressive symptoms must consider, besides gender and age group, the chronic conditions that an individual could have.

## 1. Introduction

Musculoskeletal deterioration, which is observed during the aging process, comes with severe health consequences. Even in the presence of healthy aging, there is a progressive decline in skeletal muscle quality, as described by various changes in structure, mechanics, and function [1]. Consequently, aging is accompanied by a muscle mass and strength decrease, characterised as sarcopenia [2] and dynapenia [3]. The new definition of sarcopenia by the European Working Group on Sarcopenia in Older People (EWGSOP) put muscle strength to the forefront, as it is recognised that strength is better than mass in predicting adverse outcomes [2].

Muscle weakness is commonly associated with physical frailty syndrome, besides weight loss, exhaustion, low physical activity, and slow walking speed [4]. Frailty and multimorbidity are predictors and outcomes of each other’s, as well as predictors of disability and mortality [5]. Moreover, physical frailty predicts both the onset and course of a late-life depressed mood [6]. A meta-analysis study in older adults showed that 40% of people with depression have frailty, while the prevalence of depression in frailty was equally high (38%) [7].

In addition, singly low muscle strength is associated with increased mortality, independent of the amount of physical activity, and muscle mass [8]. Among the different ways to assess strength, measuring grip strength is a relatively simple and inexpensive proxy of overall muscle strength, especially when used for clinical and epidemiological studies [9,10]. Previous studies indicate that low grip strength is associated with a variety of health outcomes, including chronic morbidities, functional disabilities, all-cause mortality [11,12], and poor mental health [13].

Chronic diseases are the leading cause of death and disability worldwide [14]. The prevalence of major chronic diseases (arthritis, heart disease, diabetes mellitus type 2, and chronic obstructive pulmonary disease) rises exponentially with age [15]. Depression is considered the second most prevalent chronic illness worldwide and is often comorbid with other chronic diseases [16]. It is estimated that depression produces the greatest decrement in health compared with other chronic diseases [17].

Several studies have reported that low grip strength is associated with depressive symptoms, in men and women, independent of age [18,19,20,21,22,23]. However, the association of grip strength and depressive symptoms in the presence of different chronic diseases has not been studied yet. Because depressive symptoms manifest different ways among different chronic diseases [17] and the prevalence of low grip strength is different among different chronic diseases [24], it is important to understand the relationship between grip strength and depressive symptoms among different chronic diseases. This information will help to further understand the possible role of muscle strength in promoting mental health in the presence of different chronic diseases. Therefore, this study aimed to explore the association between grip strength and depressive symptoms among middle-aged and older adults with different chronic diseases, while accounting for multiple confounding variables.

## 2. Materials and Methods

### 2.1. Participants and Procedures

The Survey of Health, Ageing and Retirement in Europe (SHARE) is a cross-national panel database of individual-level data on health, socio-economic status, and social and family networks of individuals aged 50 or older, from 27 European countries and Israel. More details about the project can be found elsewhere [25]. In the present study, data from wave 6 (collected in 2015) were analysed. The sample of SHARE wave 6 included 68,231 participants. In this study, the population included those who reported being clinically diagnosed with a chronic disease (rheumatoid arthritis, osteoarthritis, high blood pressure or hypertension, high blood cholesterol, diabetes, or high blood sugar) and those who reported that they did not have any chronic condition. Moreover, all participants must have: reported depressive symptoms, completed the grip strength assessment, and reported information that allowed for their characterisation (gender, age, education level, weight, height, alcohol consumption, and self-perceived health). The final sample consisted of 43,285 participants (19,925 men and 23,360 women), mean age 65.5 (SD = 10.12) from 18 countries (Austria, Germany, Sweden, Spain, Italy, France, Denmark, Greece, Switzerland, Belgium, Israel, Czech Republic, Poland, Luxembourg, Portugal, Slovenia, Estonia, and Croatia).

Face-to-face interviews to answer the questionnaires, lasting approximately 90 min, at the participant’s home, were used to collect data. The questionnaires were translated by translation experts according to the country. The first draft SHARE questionnaire was piloted with the help of the National Centre for Social Research. The pilot survey was implemented as a paper-and-pencil questionnaire in German and French samples. The Ethics Committee of the University of Mannheim, and the Ethics Council of the Max Planck Society for the Advancement of Science approved the SHARE protocol, verifying the procedures to protect confidentiality and data privacy (http://www.share-project.org/organisation/dates-facts.html). Data were collected in each country by trained researchers, following the SHARE protocol. The survey is fully described elsewhere [25,26].

### 2.2. Measures

Grip strength (kg) was assessed twice on each hand using a handgrip dynamometer (Smedley, S Dynamometer, TTM, Tokyo, Japan, 100 kg). Before the assessment, the grip strength test was explained and demonstrated, and participants had the opportunity to practice. Participants could sit or stand, with their elbow at a 90° angle, the wrist in a neutral position while keeping the upper arm tight against the trunk, and the inner lever of the dynamometer adjusted to the hand. Participants squeezed the dynamometer with their hands as hard as possible for 5 s. Values were recorded twice for each hand, alternating between left and right hands. Valid measurements were values of two measurements in one hand that differed by less than 20 kg [27]. Grip strength measurements with values of 0 kg or ≥100 kg were excluded; measurements with only grip strength recorded in one hand were also excluded. Respondents were divided into sex-specific tertiles (i.e., men: 1–40.0 kg, 41–48 kg and 48.00–99.00 kg, respectively; women: 1–25.0 kg, 26–30 kg and 31.0–99.0 kg, respectively).

Depressive symptoms were measured by the EURO-D 12-item scale. The scale details and their validation are described elsewhere [28]. Scores range between 0–12, with higher scores indicative of higher levels of depressive symptoms. Depression was analysed as a continuous variable and a dichotomic variable. For this study, a cut-off point of ≥4 points diagnoses clinically significant depression [28].

Based on a list of 14 diseases, participants were asked to report the presence or absence of each disease by disclosing whether they were previously diagnosed by a doctor. For this study, participants who received the clinical diagnosis of rheumatoid arthritis, osteoarthritis, high blood pressure or hypertension, high blood cholesterol, and diabetes or high blood sugar, or those who reported no disease, were included. These participants were divided into three groups: the ‘arthritis diseases’ group (with rheumatoid arthritis and osteoarthritis), the ‘metabolic diseases’ group (with high blood pressure or hypertension, high blood cholesterol, and diabetes or high blood sugar), and the ‘no disease’ group (if participants did not have a clinical diagnosis).

Potential confounders included gender, age, education level, living place, alcohol consumption, body mass index (BMI), and self-perceived health. Studies already have proven that there are significant differences in the association of grip strength and depressive symptoms between gender and age [20,23]. Educational level and living place are associated to both grip strength and depression [29]. Alcohol consumption and BMI were previously used as confounder variables in similar analyses [20,22,23]. Self-reported health is an important variable when the sample is comprised of older adults, especially considering participants with chronic disease, as the subjective evaluation of health could be a proxy of the individual health status influencing both grip strength and mental health [30].

Age was recorded into three groups: 50–64, 65–79, and ≥80 years. Education level was grouped as low, middle, or high, according to the International Standard Classification of Education 1997 (ISCED-97). For living place, participants reported if they lived in a big city, the suburbs or outside of a big city, a large town, a small town or in a rural area or village. Alcohol consumption was dichotomised into ‘Yes’ for those who reported drinking six or more alcoholic drinks monthly, and ‘No’ for those who reported drinking less than six drinks per month. BMI was calculated from self-reported height and weight, dividing the weight (kg) by the square of height (m). Self-perceived health provides information concerning how an individual perceives his/her health, rating it as very bad, bad, fair, good, or very good. The question was, ‘How is your health in general?’ This single-item question has been widely validated in epidemiological studies [31,32]. The validity was previously tested in studies that found a relationship between levels of self-perceived health and adverse health outcomes [33].

### 2.3. Data Analysis

Statistical analyses were conducted using SPSS version 25.0. Descriptive statistics for all variables were calculated (percentage; mean and standard deviation; median and minimum and maximum). The analyses were stratified into three groups: no disease, arthritis diseases, and metabolic diseases. Multivariate binary logistic regressions were conducted to assess the association between grip strength and depression. The first model was adjusted for gender; the second model was adjusted for gender, age, country, education level, living place and BMI; and third model was further adjusted for self-perceived health. Statistical significance was set as two-sided *p* < 0.05.

## 3. Results

Participants’ characteristics are provided in Table 1. The overall sample was 46% men and 54% women, mean age is 65.5 (10.1). The highest mean EURO-D depression score was 2.64 (95% confidence interval (CI): 2.58 to 2.71) for the ‘arthritis diseases’ group, followed by 2.16 (95% CI: 2.13 to 2.19) for the ‘metabolic diseases’ group and, finally, 1.84 (95% CI: 1.81 to 1.86) for the ‘no disease’ group. The inverse order was observed for grip strength. The highest value 35.68 (95% CI: 35.52 to 35.83) was for the ‘no disease’ group, followed by 34.50 (95% CI: 34.34 to 34.67) for the ‘metabolic diseases’ group and 30.91 (95% CI: 30.57 to 31.25) for the ‘arthritis diseases’ group. A significant statistical difference was observed between all groups and for both depressive symptoms and grip strength (*p* < 0.001).

Table 2 presents the odds ratios (OR) for the association between relative grip strength and risk of depression among the ‘no disease’ group, the ‘arthritis diseases’ group, and the ‘metabolic diseases’ group. The results, for all groups, showed that grip strength was inversely associated with depression. Model 2 results, adjusted for gender, age, country, education, living place, alcohol consumption, and BMI are summarised here. Compared to those in the low strength tertile, those in the middle and high strength tertiles, in the ‘no disease’ group, had a statistically significant 35% (*p* < 0.001) and 34% (*p* < 0.001) lower odds of depression, respectively. For those in the ‘arthritis diseases’ group, compared to those in the low strength tertile, those in the moderate and high strength tertiles had a significant 31% (*p* = 0.032) and 35% (*p* < 0.005) lower odds of depression, respectively. In the ‘metabolic diseases’ group, compared to those in the low strength tertile, those in the moderate and high strength tertiles had statistically significant 41% (*p* < 0.001) and 44% (*p* < 0.001) lower odds of depression, respectively. Model 3 was adjusted for all variables of model 2 and additionally for self-perceived health. With this adjustment, significant associations found in the ‘arthritis diseases’ group for model 2 were lost, and thus no difference between grip strength levels was found regarding depression. For the ‘no disease’ and the ‘metabolic diseases’ groups, the significant association between grip strength and depression remained. Thus, in the ‘no disease’ group, compared to those with low grip strength, those with moderate and high strength had 25% (*p* = 0.011) and 18% (*p* < 0.001) lower odds of depression, respectively. Similarly, in the ‘metabolic diseases’ group, those in the moderate and higher strength tertiles, had 31% (*p* < 0.001) and 30% (*p* < 0.001) lower odds of depression, respectively, compared to those in the lower strength tertile.

## 4. Discussion

The purpose of this study was to investigate the association between grip strength and depressive symptoms, considering different chronic diseases in European citizens over the age of 50. Data came from a transnational population-based study, among 16 European countries. It was found, in model 1 (adjusted for gender) and model 2 (adjusted for gender, age, country, education level, living place, drinking alcohol, and body mass index), that higher grip strength is significantly associated with having lower odds of depression, within all three chronic disease groups. However, when the analyses were adjusted for model 2 and self-perceived health, the association was attenuated and became non-significant in individuals with arthritis diseases, osteoarthritis and rheumatoid arthritis. No difference was observed to be in the moderate or in the hight tertile for all groups analysed. Although a significant inverse association between grip strength and depressive symptoms was consistently found in previous research [19,20,21,22,23,24,34], for the first time, to the best of our knowledge, this association was analysed considering the fact of having or not having a chronic disease (and the type of this disease).

In our study, comparing the three groups (‘no disease’, ‘arthritis’, and ‘metabolic’), the arthritis group presents the lowest values for the variables grip strength and depressive symptoms. Other studies analysed the positive relationship between arthritis and depression [35,36], and arthritis and low grip strength [37]. Another important fact, that must be analysed in this study, is that the significant association between grip strength and depressive symptoms ceased to exist for individuals with arthritis diseases when the model was adjusted for the self-perceived health variable (in addition to gender, age, country, education, living place and BMI). High rates of chronic illness, mental health conditions, disability and frailty may reduce self-perceived health. In Europe, 41.4% of people 65 years and over evaluate their health as good or very good [38]. The influence on perceived health by chronic conditions would be, to some extent, explained by their disabling capacity, rather than only by direct effect [39]. In our study, 45.7% of individuals in the no disease group evaluated their health as very good and excellent, while, for the metabolic disease group, this value decreased to 22.6%. In the arthritis diseases group, this value decreased even more—only 18.8% of individuals evaluated their health as very good and excellent. Arthritis diseases are very disabling diseases and have a huge impact on an individual’s life. Perceived health, as an approximation of the overall health status, may influence both grip strength and depressive symptoms.

Understanding the causal mechanism underlying the relationship between grip strength and depressive symptoms could be the key to understanding why this association may be different according to various chronic diseases. One possibility is that depression may cause a decline in systematic physical functioning through its association with adverse health behaviours [40]. Moreover, decreased functional performance results in reduced ability to undertake one’s activities of daily living, which increases social isolation and the risk of depression [41]. The relationship between depressive symptoms and grip strength is likely bidirectional [20]. However, it seems that this relationship should be considered separately for those with different chronic conditions. In an American sample, from the National Health and Nutrition Examination Surveys, no significant difference in depressive symptoms was observed in individuals with diabetes or obesity compared to those with neither condition, although participants reported significantly more difficulty with physical function [42]. In a study with osteoarthritis patients, a significant association was found between depression and self-reported measures of function [43].

Furthermore, sarcopenia has been related to increased peripheral inflammation [44], which is closely related to the underpinning mechanisms of depression [45]. The coexistence of immune-mediated inflammatory diseases (e.g., rheumatoid arthritis) with depression has long been recognised [46]. Depression and arthritis coexist more often than would be predicted by chance [47], which has also been noted for several other somatic conditions associated with systemic inflammatory responses, including cardiovascular disease [48], diabetes [49] and obesity [50]. These findings suggest that other factors may be at play, and need to be examined. One of them may be the brain-derived neurotrophic factor (BDNF) which is negatively correlated with depression [51] and its level is also lower in rheumatoid arthritis [52].

Skeletal muscles can express several ‘myokines’ into circulation in response to muscle contractions. BDNF could be considered as a novel contraction-induced myokine [53]. The anti-inflammatory and anti-atherogenic properties of these myokines protect against the risk of depression [45]. Myokine is recognised as a major contributing factor that links muscular activity with health [54]. Studies had been proven that physical exercise can increase circulating BDNF levels [55,56]. In this sense, low grip strength could be associated with low BDNF.

Moreover, between skeletal muscles and the brain there is a crosstalk. Physical activity could activate many cell pathways in muscles and in the brain. Myokines, among others, BDNF, could cross the blood–brain barrier [57]. BDNF is also a neurokine. Under physical activity, it is secreted in the different regions of the brain in large amounts and triggers its biological effects by specific receptors. It could also cross the blood–brain barrier and act peripherally.

BDNF is also one of the factors involved in inflammatory process taking part in the neuroimmune axis regulation, together with cytokines and chemokines. Jin and Sun et al. [58] report that expression of BDNF is strongly affected by immune cells and the immune factors they secrete. Dysregulation of cytokine levels in the brain intensifies inflammation and increases reactive oxygen species production, which disrupts neuronal homeostasis and neurogenesis [59,60].

In addition to these common associations of depression with a variety of chronic conditions, the present study reveals that the highest value of depression is in the ‘arthritis diseases’ group; a significant difference, when comparing the ‘no disease’ group with the ‘metabolic diseases’ group, was observed (*p* < 0.001).

The current findings should be interpreted in light of some limitations. The study is cross-sectional; therefore, the directionality of the relationships between grip strength and depressive symptoms cannot be deduced with certainty. Longitudinal studies are required to better disentangle the relationships we observed. There was no information about participants’ hand sizes and no consideration of the cultural variation in the use of grip strength in daily social routines. There was a lack of chronic disease details, like how long it had been since the individuals received the diagnosis, the disease severity, and the impairment related to the disease. Moreover, we did not know if participants were receiving pharmacological treatments (i.e., antidepressant, anti-inflammatory, and/or antidiabetic, antihypertensive drugs), which could interfere in the association of grip strength and depressive symptoms. Finally, the analyses were done by considering the total sample, but did not considered gender difference. For future study, it will be interesting to analyse this association by considering both chronic disease groups and gender. Nonetheless, the strengths of the study include a large sample from several countries, which enables us to assess the relationship between grip strength and depressive symptoms in the European population and provide clearer estimates of the effects, and objectively measure grip strength.

## 5. Conclusions

Our data provide a platform to investigate whether preserving a sufficient level of strength can reduce the risk of depression among middle-aged and older adults (considering an individual’s chronic condition). We found that the association of grip strength with depressive symptoms is significant and inverse in individuals without chronic conditions and in individuals with metabolic diseases. In individuals with arthritis disease, this association is not significant when the model is adjusted for self-perceived health. The influence of self-perceived health on the association of grip strength and depression should be investigated in future studies. This study reinforces the requirement to understand the correlation between physical and mental health.

## Figures and Tables

**Table 1 ijerph-17-06942-t001:** Participants’ characteristics (*n* = 43,285).

Variables	Overall 43,285 (%)	No Disease 20,520 (47.4%)	Arthritis Diseases 4197 (9.7%)	Metabolic Diseases 18,568 (42.9%)	*p* Value
**Gender**					<0.001
Men	19,911 (46.0)	9254 (45.1)	1343 (32.1)	9321 (50.2)	
Women	23,374 (54.0)	11,266 (54.9)	2854 (67.9)	9247 (49.8)	
**Age**	65.5 (10.12)	63.3 (9.94)	66.8 (10.96)	67.6 (9.61)	<0.001
**Age group**					<0.001
50–64 years	21,470 (49.6)	12,291 (59.9)	1876 (44.7)	7334 (39.5)	
65–79 years	17,660 (40.8)	6792 (33.1)	1767 (42.1)	9098 (49.0)	
>80 years	4155 (9.6)	1437 (7.0)	554 (13.2)	2136 (11.5)	
**Education level**					<0.001
Low	15,669 (36.2)	6505 (31.7)	1591 (37.9)	7743 (41.7)	
Middle	16,838 (38.9)	8228 (40.1)	1645 (39.2)	6889 (37.1)	
High	10,778 (24.9)	5787 (28.2)	961 (22.9)	3936 (21.2)	
**BMI**	26.71 (4.34)	25.73 (3.94)	26.45 (4.42)	27.8 (4.46)	<0.001
**Alcohol ≥6 drinks**					0.007
Yes	1039 (2.4)	513 (2.5)	88 (2.1)	446 (2.4)	
No	42,246 (97.6)	20,007 (97.5)	4109 (97.9)	18,122 (97.6)	
**Self-perceived health**					<0.001
Poor	1991 (4.6)	533 (2.6)	331 (7.9)	1114 (6.0)	
Fair	9696 (22.4)	3119 (15.2)	1360 (32.4)	5237 (28.2)	
Good	17,271 (39.9)	7531 (36.7)	1717 (40.9)	8021 (43.2)	
Very good	10,129 (23.4)	6218 (30.3)	642 (15.3)	3268 (17.6)	
Excellent	4198 (9.7)	3119 (15.2)	147 (3.5)	928 (5.0)	
**EURO-D scale (score)**	2.00 (0–12)	1.00 (0–12)	2.00 (0–11)	2.00 (0–12)	<0.001
**Depression**					<0.001
No	34,325 (79.3)	16,971 (82.7)	2908 (69.3)	14,427 (77.7)	
Yes	8960 (20.7)	3549 (17.3)	1289 (30.7)	4141 (22.3)	
**Grip strength (kg)**	34.71 (11.63)	35.68 (11.52)	30.91 (11.28)	34.50 (11.65)	<0.001

Data are *n* (%), mean (standard deviation) for age; and median (minimum and maximum) for EURO-D scale (score); BMI, body mass index; *p*-values were calculated with chi-squared test and one-way ANOVA test for categorical and continuous variables respectively.

**Table 2 ijerph-17-06942-t002:** Logistic regression models for the association of relative grip strength with depression by disease group.

Grip Strength Level	Model 1 OR (95% CI) ^1^	Model 2 OR (95% CI) ^2^	Model 3 OR (95% CI) ^3^
**No disease (*n* = 20,520)**			
Low (*n* = 6217)	1.00 (REF)	1.00 (REF)	1.00 (REF)
Moderate (*n* = 7182)	0.69 (0.63 to 0.75)	0.65 (0.58 to 0.73)	0.75 (0.66 to 0.84)
High (*n* = 7121)	0.67 (0.62 to 0.73)	0.66 (0.58 to 0.74)	0.82 (0.72 to 0.93)
**Arthritis diseases (*n* = 4197)**			
Low (*n* = 1989)	1.00 (REF)	1.00 (REF)	1.00 (REF)
Moderate (*n* = 1276)	0.69 (0.58 to 0.81)	0.69 (0.56 to 0.85)	0.86 (0.69 to 1.06)
High (*n* = 932)	0.63 (0.53 to 0.75)	0.65 (0.52 to 0.82)	0.88 (0.69 to 1.12)
**Metabolic diseases (*n* = 18,568)**			
Low (*n* = 6424)	1.00 (REF)	1.00 (REF)	1.00 (REF)
Moderate (*n* = 5905)	0.63 (0.58 to 0.69)	0.59 (0.53 to 0.66)	0.69 (0.61 to 0.77)
High (*n* = 6239)	0.59 (0.54 to 0.64)	0.56 (0.50 to 0.64)	0.71 (0.63 to 0.81)

OR, odds ratio; CI, confidence interval; REF, reference value; ^1^ adjusted for gender; ^2^ adjusted for gender, age, country, education level, living place, drinking alcohol, and body mass index; ^3^ adjusted for model 2 and self-perceived health.

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
