# Peer review of "The Association of Grip Strength with Depressive Symptoms among Middle-Aged and Older Adults with Different Chronic Diseases"

_ijerph, 2020, doi:10.3390/ijerph17196942_

Round 1

Reviewer 1 Report

I feel that the authors adequately addressed my comment and, as far as I am concerned, I feel that the manuscript is suitable for publication.

Author Response

Reviewer 1

Comment: I feel that the authors adequately addressed my comment and, as far as I am concerned, I feel that the manuscript is suitable for publication.

Response: Thank you. Your contribution was very important to improve our paper.

Reviewer 2 Report

Spelling check: 

Discussion: Miokine, amoung, excistance, secration etc

The paragraph on several chemokines and hormones in the discussion section is in complete incoherence with the study wherein the authors have not measured the levels of the hormones in their participants but have rather just mentioned them.. It is difficult to determine the merit of those hormones in the study being so unrelated to the 'purpose' of the study.

Author Response

Reviewer 2

Comment: Spelling check: Discussion: Miokine, amoung, excistance, secretion etc.

Response: Sorry for these spelling mistakes. We did the correction.

Comment: The paragraph on several chemokines and hormones in the discussion section is in complete incoherence with the study wherein the authors have not measured the levels of the hormones in their participants but have rather just mentioned them. It is difficult to determine the merit of those hormones in the study being so unrelated to the 'purpose' of the study.

Response: Thank you for you observation. We reflect and agree with you. So, we decided to remove this paragraph.

Reviewer 3 Report

Dear authors,

thank you for considering my further suggestions and therefore improving the quality of the paper.

Author Response

Reviewer 3

Comment: Thank you for considering my further suggestions and therefore improving the quality of the paper.

Response: Thank you for your valuable contribution.

Reviewer 4 Report

The major issue in this study is the methods used to collect data:

  1. How was the questionnaire validated after translation?
  2. According to the methods section, it seems that the researchers have used both questionnaire and interview. Which one? And how were the interviews analysed?
  3. What was the number of participants conducted interview?
  4. What was the purpose of the interviews?
  5. The link provided in the method section regarding “protect confidentiality and data privacy” is not working?

Author Response

Reviewer 4

Comment: How was the questionnaire validated after translation?

Response: Each country participating in the SHARE Project was responsible for the questionnaire validation process.

Comment: According to the methods section, it seems that the researchers have used both questionnaire and interview. Which one? And how were the interviews analysed?

Response: Sorry about the misunderstood. The Share Project use interview to aswer the questionnaires. The questionnaires were not self-fill. In line 88 we add to the sentence this explanation to clarify: “Face-to-face interviews to answer the questionnaires”.

Comment: What was the number of participants conducted interview?

Response: The sample of SHARE wave 6 included 68231 participants. In our study, the population included those who reported being clinically diagnosed with a chronic disease (rheumatoid arthritis, osteoarthritis, high blood pressure or hypertension, high blood cholesterol, diabetes, or high blood sugar) and those who reported that they did not have any chronic condition. Moreover, all participants must have: reported depressive symptoms, completed the grip strength assessment, and reported information that allowed for their characterization (gender, age, education level, weight, height, alcohol consumption, and self-perceived health). The final sample of our study, consisted of 43285 participants.

This information is on the Materials and Methods section.

Comment: What was the purpose of the interviews?

Response: To facilitate understanding of the questionnaires.

Comment: The link provided in the method section regarding “protect confidentiality and data privacy” is not working?

Response: Sorry about that. We retest the link and it is ok. You also can found more information in: http://www.share-project.org/home0.html.

This manuscript is a resubmission of an earlier submission. The following is a list of the peer review reports and author responses from that submission.

Round 1

Reviewer 1 Report

The present study by Marconcin and collaborators explores the putative relationship between grip strength and major depression in a context of comorbidity with chronic diseases such as Arthritis diseases or metabolic disorders. One of the most remarkable results reported herein is the fact that that higher grip strength is significantly associated with having lower odds of depression, independent of having or not having a chronic disease. This study is interesting since such association is poorly documented and is particularly relevant to highlight easily identifiable predictive markers of mental health disorders.

I have only minor comments that can be used by the authors to improve their manuscript on specific aspects.

1)        The introduction should introduce the notion of frailty and how this parameter is a predictor of functional disabilities but also mood disorders notably during aging.

2)        The link between depression and grip strength is merely introduced and, according to the authors, relies on association studies (refs 14-19). Mechanistic hypotheses are proposed in the discussion (ie, peripheral inflammation, BDNF...) but I feel that these considerations should be strengthened. In particular what is the role of the monoaminergic systems in grip strength. The notion of BDNF is also interesting when studying depression. However, this is not sufficiently addressed in the discussion. For example, BDNF is produced by the brain but also by muscles and considered as a myokine. This could be further discussed.

3)        Regarding the metabolic diseases, this is a generic term that includes several pathologies : “high blood pressure or hypertension, high blood cholesterol, diabetes or high blood sugar » with distinct mechanisms of action. This complicates the identification of the causal mechanisms.

4)        The EURO-D 12-item scale was used to score depressive state. Is there any advantage to use this scale compared to conventional scales such as Hamilton rating scale or MADRS? Would the results/conclusions be the same using the latter scales?

5)        The reviewer is wondering whether or not patients included in these studies are under pharmacological treatments (ie, antidepressant, anti-inflammatory and/or antidiabetic, antihypertensive drugs). If yes, how can the authors dissociate the effect of the pathological state to that of the pharmacological interventions? Is it possible to adjust th results taking this variable into consideration?

6)        Finally, in their discussion the authors should be more persuasive about the interest to associate grip strength with depressive-symptoms whatever the comorbidity.  

Author Response

The present study by Marconcin and collaborators explores the putative relationship between grip strength and major depression in a context of comorbidity with chronic diseases such as Arthritis diseases or metabolic disorders. One of the most remarkable results reported here in is the fact that higher grip strength is significantly associated with having lower odds of depression, independent of having or not having a chronic disease. This study is interesting since such association is poorly documented and is particularly relevant to highlight easily identifiable predictive markers of mental health disorders. I have only minor comments that can be used by the authors to improve their manuscript on specific aspects.

Comment: The introduction should introduce the notion of frailty and how this parameter is a predictor of functional disabilities but also mood disorders notably during aging.

Response: Thank you for your comment. Frailty is an important syndrome related to both the aging process and muscle weakness. We agree that it is relevant to be mentioned in the introduction. We added one paragraph with this topic and three relevant references.

Comment: The link between depression and grip strength is merely introduced and, according to the authors, relies on association studies (refs 14-19). Mechanistic hypotheses are proposed in the discussion (ie, peripheral inflammation, BDNF...) but I feel that these considerations should be strengthened. In particular, what is the role of the monoaminergic systems in grip strength. The notion of BDNF is also interesting when studying depression. However, this is not sufficiently addressed in the discussion. For example, BDNF is produced by the brain but also by muscles and considered as a myokine. This could be further discussed.

Response: Thank you for your contribution. We addressed the role of exercise and consequently, muscle function in increasing the BDNF factor. We added at the discussion section four paragraphs to clarify this mechanism.

Comment: Regarding the metabolic diseases, this is a generic term that includes several pathologies: “high blood pressure or hypertension, high blood cholesterol, diabetes or high blood sugar” with distinct mechanisms of action. This complicates the identification of the causal mechanisms.

Response: We agree with your comment, but high blood pressure or hypertension, high blood cholesterol, diabetes, or high blood sugar had a similar correlation with depressive symptoms and handgrip variables. But for future studies, it will be interesting to investigate the pathologies separated.

Comment: The EURO-D 12-item scale was used to score depressive state. Is there any advantage to use this scale compared to conventional scales such as Hamilton rating scale or MADRS? Would the results/conclusions be the same using the latter scales?

Response: Its brevity and simplicity allow the use as a screening instrument and as an outcomes measure for health services research. EURO-D is a reliable instrument such as the Hamilton rating scale, MADRS, or CES-D.

Comment: The reviewer is wondering whether or not patients included in these studies are under pharmacological treatments (ie, antidepressant, anti-inflammatory and/or antidiabetic, antihypertensive drugs). If yes, how can the authors dissociate the effect of the pathological state to that of the pharmacological interventions? Is it possible to adjust the results taking this variable into consideration?

Response: We did not have this information. So, we added this information as a limitation of the study in the discussion section.

Comment: Finally, in their discussion the authors should be more persuasive about the interest to associate grip strength with depressive-symptoms whatever the comorbidity.

Response: Thank you. We improved the discussion section to reinforce this idea.

Reviewer 2 Report

Minor spelling corrections like recorded instead of recoded in line 110 etc

Could the authors explain the correlation between the level of education and grip strength as a parameter of the study?

As the N number is quite high, the study is ought to give significant p values, however, the reason for the correlation between arthritis and depression is quite eclipsed in the current study. Eg: Could the study better explain the part of the brain that get affected during depression and its correlation to bone health perhaps, if it exists?

Author Response

Comment: Minor spelling corrections like recorded instead of recoded in line 110 etc

Response: Thank you and sorry about this lack of attention. We proceeded with the correction of the spelling.

Comment: Could the authors explain the correlation between the level of education and grip strength as a parameter of the study?

Response: Thank you. A paragraph explaining this correlation and also the correlation of the other confounders variables was added to the methods section.

Comment: As the N number is quite high, the study is ought to give significant p values, however, the reason for the correlation between arthritis and depression is quite eclipsed in the current study. Eg: Could the study better explain the part of the brain that get affected during depression and its correlation to bone health perhaps, if it exists?

Response: Thank you for your observation. We added in the discussion section an explanation about the relation of brain-derived neurotrophic factor (BDNF) and depression. As BDNF is produced by muscle cells and is strong and negatively associated with depression, it could explain this association. Moreover, we mentioned that depression may cause a decline in systematic physical functioning through its association with adverse health behaviours. Also, when functional performance decrease it results in a reduction of the ability to undertake one's activities of daily living, which increases social isolation and risk of depression.

Reviewer 3 Report

This study emphasizes the association of grip strength with depressive symptoms among older adults in the presence of different chronic comorbidities. This manuscript addresses an important area to further understand the association of physical activity and mental illness. Another strength of the study is the large international sample. Nevertheless, I have a number of suggestions outlined below.

Headline: Maybe you could use the phrase "depressive symptoms or symptoms of depression" instead of "depression symptoms". Why is the target group of both adults and old adults mentioned in the title? As your sample is 50 years or older, maybe use the term "older people/adults".

L25: Can you revise this sentence? And p-Value should be reported as p<.001 (please consider this for the whole manuscript)

L42: Can you add "amount of" physical activity

L58: Can you elaborate, why controlling for chronic diseases is important?

L81: Who translated the questionnaire? Can you give details on the pilot phase?

L92: Is there a paper you refer to for valid measurement?

L 127: Can you give details on why and how you chose these two models?

L140: please add "n" for the different variables

L147: please add "n" for the different variables

L160: please add "n" for the different variables

L147: Table 2: Can you give details on why and how gender differences contribute to the results of your work?

L166: "self perceived"

L168: Please find a more appropriate expression for "on the other hand"

L199: 45,7% "of" individuals

L201: Please consider revising this sentence

L209: Word repetition, please revise this sentence.

L232: Please consider revising this sentence

L242: Maybe you could write "to understand the correlation of physical and mental health"

Author Response

Comment: This study emphasizes the association of grip strength with depressive symptoms among older adults in the presence of different chronic comorbidities. This manuscript addresses an important area to further understand the association of physical activity and mental illness. Another strength of the study is the large international sample. Nevertheless, I have a number of suggestions outlined below.

Response: Thank you.

Comment: Headline: Maybe you could use the phrase "depressive symptoms or symptoms of depression" instead of "depression symptoms".

Response: Thank you for the observation. We decided to change in the role paper this expression.

Comment: Why is the target group of both adults and old adults mentioned in the title? As your sample is 50 years or older, maybe use the term "older people/adults".

Response: Thank you for your suggestion. The editor had the same opinion. We changed and used the terminology “middle-aged and older adults”.

Comment: L25: Can you revise this sentence? And p-Value should be reported as p<.001 (please consider this for the whole manuscript)

Response: The sentence was revised and the reported p-value was changed in the manuscript.

Comment: L42: Can you add "amount of" physical activity

Response: Yes. We added “amount of”, as suggested.

Comment: L58: Can you elaborate, why controlling for chronic diseases is important?

Response: We review this paragraph and added an explanation.

Comment: L81: Who translated the questionnaire? Can you give details on the pilot phase?

Response: The questionnaires were translated by translation experts. In the text, we added a bit about the pilot phase.

Comment: L92: Is there a paper you refer to for valid measurement?

Response: Yes, the reference was added after the sentence.

Comment: L 127: Can you give details on why and how you chose these two models?

Response: Actually, we chose three models. First, we adjusted only for gender. Model two were adjusted for gender, age, education, living place, BMI, and self-perceived health. We now included one paragraph on the methods section to justify why these variables are important in the association of grip strength and depressive symptoms. Then, in model 3, we included the variable self-perceived health.  We considered self-perceived health as a proxy of the health status and therefore a confounder of the association between grip strength and depression. We decided to present three models for the reader to see these differences.

Comment: L140: please add "n" for the different variables

Response: We added n for each variable.

Comment: L147: please add "n" for the different variables

Response: We decided to remove this table from the paper.

Comment: L160: please add "n" for the different variables

Response: We added n for each variable.

Comment: L147: Table 2: Can you give details on why and how gender differences contribute to the results of your work?

Response: We did not analyse the association between grip strength and depressive symptoms considering sex differences. We decided to remove the table and paragraph with these analyses from the results section.

Comment: L166: "self perceived"

Response: Thank you for the correction.

Comment: L168: Please find a more appropriate expression for "on the other hand"

Response: The expression was withdrawn. The text became clearer and more objective, thank you.

Comment: L199: 45,7% "of" individuals

Response: Thank you for the correction.

Comment: L201: Please consider revising this sentence

Response: We rewrite the sentence to be more understandable.

Comment: L209: Word repetition, please revise this sentence.

Response: Thank you for the correction.

Comment: L232: Please consider revising this sentence

Response: The sentence was revised.

Comment: L242: Maybe you could write "to understand the correlation of physical and mental health"

Response: Thank you for your suggestion.

Reviewer 4 Report

This study investigate “the association of grip strength with depression symptoms among adults and old adults in different chronic diseases”. Nevertheless, the introduction is very unscientific, lack flow and structure. The method section is very confusing and missing great deal of detail. E.g., what was the methodological design of the study? Not stated. Who accessed Grip strength (the authors)?, how was it possible to interview 43285 persons between the year 2015 until this day?, who conducted the interviews? The method part of this study is very unclear. How was the ethical approved on data that the authors have not been part of collecting as far as I understand? Where can we find the ethical approval of this study? This study has a major ethical concerns.

Author Response

Comment: This study investigate “the association of grip strength with depression symptoms among adults and old adults in different chronic diseases”. Nevertheless, the introduction is very unscientific, lack flow and structure. The method section is very confusing and missing great deal of detail. E.g., what was the methodological design of the study? Not stated. Who accessed Grip strength (the authors)?, how was it possible to interview 43285 persons between the year 2015 until this day?, who conducted the interviews? The method part of this study is very unclear. How was the ethical approved on data that the authors have not been part of collecting as far as I understand? Where can we find the ethical approval of this study? This study has a major ethical concerns.

Response: Thank you for your comment. With your feedback and also from the other reviewers, we improved our paper significantly. In the introduction section, we now added an input regarding frailty syndrome and reinforced the importance of analyse the association of grip strength and depressive symptoms taking into consideration chronic diseases. Also, the methods section was enhanced to address your concerns. The ethical approval can be found at http://www.share-project.org/organisation/dates-facts.html. The discussion section has undergone improvements too, we add more detailed explanations concerns the role of BDNF protein. I hope these enhancements meets your expectation regarding our paper.

Round 2

Reviewer 2 Report

The current version seems way better than the previous one. 

Reviewer 3 Report

Thank you for adressing the comments and concerns I have made. The manuscript has been improved, however I have some comments I would like to address. Especially, english language and style are not appropiate. Furthermore, the discussion lacks of a thought out argumentation and a well-reasoned conclusion.

 L 121: Studies already have proven that there are significant...

L 123: ...are associated to both...

L 205: Please rewrite the sentence. Suggestion: "Interestingly, there was only a significant difference between the medium and high grip strength level for the no disease group in association with depression." The next sentence is redundant. I wouldn't agree to the last sentence of this paragraph. You use the word "medium" in the figure and "moderate" in the text, please choose one expersion.

L211: Can you use "lowest" instead of "worst"

L225: Please rewrite the sentence. How can a relationship between grip strength and depressive symptos consider self-perceived health?

L297: Please rewrite the sentence.

L299: Suggestion: The influence of self-perceived health on the association of grip strenght and depression should be investigated in future studies, to understand the correlation between pa and mh.